# Nicotine and Nicotine-Free Vaping Behavior among a Sample of Canadian High School Students: A Cross-Sectional Study

**DOI:** 10.3390/children10020368

**Published:** 2023-02-13

**Authors:** Evan R. Wiley, Jamie A. Seabrook

**Affiliations:** 1Department of Epidemiology and Biostatistics, Western University, London, ON N6A 3K7, Canada; 2School of Food and Nutritional Sciences, Brescia University College, London, ON N6G 1H2, Canada; 3Department of Pediatrics, Western University, London, ON N6A 3K7, Canada; 4Children’s Health Research Institute, London, ON N6C 2V5, Canada; 5Lawson Health Research Institute, London, ON N6C 2R5, Canada

**Keywords:** vaping, e-cigarette use, nicotine vaping, nicotine-free vaping, epidemiology

## Abstract

Youth vaping is a public health concern in Canada. Researchers have explored factors associated with vape use, but rarely differentiated between types of use. This study estimates the prevalence and correlations among past-month nicotine vaping, nicotine-free vaping, and dual-use vaping (nicotine and nicotine-free) in grades 9–12 high school students. Data came from the 2019 Canadian Student Tobacco, Alcohol and Drugs Survey (CSTADS). The total sample consisted of 38,229 students. We used multinomial regression to assess for the correlations among different categories of vape use. Approximately 12% of the students reported past-month vape use exclusively with nicotine, 2.8% reported exclusively nicotine-free vape use, and 14% reported both nicotine vaping and nicotine-free vaping. Substance use (smoking, alcohol, cannabis) and being male were associated with membership in every category of vape use. Age was associated with vape use, but in different directions. Grade 10 and 11 students were more likely than grade 9 students to vape exclusively with nicotine (aOR 1.36; 95% CI: 1.05, 1.77 and aOR 1.46; 95% CI: 1.09, 1.97), while grade 9 students were more likely than grade 11 and 12 students to vape with both nicotine and nicotine-free vapes (aOR 0.82; 95% CI: 0.67, 0.99 and aOR 0.49; 95% CI: 0.37, 0.64). The prevalence of nicotine and nicotine-free vaping is high, with many students reporting the use of both.

## 1. Introduction

Adolescence represents a distinct period marked by social and developmental changes. With a shift in social norms and pressure to fit in, coupled with developmental changes in the limbic system and the prefrontal cortex, risky behaviors—including the initiation of substance use—tend to increase [1,2]. Well-established research has explored risk factors and trajectories associated with adolescent alcohol, tobacco, and (to a lesser extent) cannabis use, but much less is known about vaping [3,4].

Vapes, also known as electronic cigarettes, e-cigarettes, and electronic-nicotine-delivery systems (ENDS), have continued to grow in popularity [5,6,7]. A vaping apparatus creates a vapor out of a liquid that is frequently flavored (e.g., candies, fruits, desserts) and may contain nicotine or tetrahydrocannabinol (THC). The vapor is then inhaled. These vaping devices have become very popular among non-smoking youth [6]. Recent estimates suggest that nearly 15% of Canadians have tried vaping at some point in their lives, with a disproportionate level of use occurring among adolescents and high school students [8,9,10]. In fact, between 2017 and 2019, past-week vape use more than doubled among adolescents aged 16–19 years in Canada [6]. In 2019, 25.7% of Ontario high school students reported current use [5]. The high prevalence of youth vaping is largely attributable to aggressive marketing, flavored vape products, and reactionary legislation [11].

While the long-term health effects of vaping remain unknown, recent research suggests that adolescent vaping is associated with a series of negative physical and mental health outcomes, including nicotine addiction and exposure to harmful chemicals [8], as well as seizures from nicotine toxicity, acute pulmonary injury, and explosions with resultant burns and blast effects [11]. In addition, adolescent vaping has been implicated in multiple patterns of polysubstance use. Seabrook et al. [7] found that alcohol, tobacco, cannabis, and cocaine use were all associated with greater odds of past-month vape use, while Zuckermann et al. [12] found that vaping substances were commonly combined with other substances. 

Other research has explored how adolescent vaping can influence the likelihood of initiating cigarette smoking and marijuana use [13,14,15,16], although it is unclear whether there is a threshold of vape use that is predictive of other substance use [1]. Dai and Hao [13] found that among youth who had never smoked cigarettes, those who used a flavored vape in the past month had a nearly six-fold increase in the intention to initiate cigarette smoking compared to those who had not vaped in the past month, and 40% lower odds of intention to quit cigarette smoking among current smokers. In an analysis of prospective nationally representative longitudinal data from the U.S., Miech et al. [14] found that vaping strongly predicted cigarette smoking one year later among grade 12 students who had never previously smoked cigarettes. The authors speculated that youth who vape for experimentation purposes may not foresee any immediate consequences as a result of their vaping behavior and may wrongly conclude that the dangers of cigarette smoking are also exaggerated, especially among those who have never smoked. Similarly, in their longitudinal study, Dai et al. [16] found that vape use predicted future marijuana use among youth. In a community sample of 13–17-year-old adolescents from Erie County, NY, Park et al. [1] found that both high and low levels of vape use were correlated with higher alcohol and cannabis use over time, suggesting the need for early intervention and prevention of vape use altogether. In addition to the risks associated with other substance use, researchers have explored connections between adolescent vape use and mental health. Baiden et al. [17] and Welty et al. [18] found that current adolescent vape users were more likely to suffer from a variety of mental health issues, including depressive symptoms, suicidal ideation, suicidal plans, and attempting suicide. The associations also appear to be stronger for adolescent females than for males [17].

Although research related to adolescent vaping continues to expand, most studies have neglected an important distinction—i.e., different types of vape users. A common approach has been to treat vape use as a unitary behavior, even though some users vape with nicotine products while others do not. Moreover, there may be important differences between youth who vape with and without nicotine [19,20]. Tokle et al. [20] found that those who vaped with nicotine had more behavioral and mental health issues than those who vaped nicotine-free. Other research has shown that self-reported use of vaping products with higher nicotine concentrations positively correlates with vaping frequency and perceived vaping addiction, and that nicotine addiction could, in turn, impact brain development [6]. Nevertheless, despite the presence of different types of vape users, little is known about health-related and/or sociodemographic differences between nicotine and nicotine-free vape users. In 2021, the Canadian Public Health Association echoed this sentiment, issuing a report that explicitly advocated for future research in the area of nicotine and nicotine-free vape use [21].

Given the high and growing prevalence of adolescent vape use, coupled with the lack of research on specific types of vape users, including risk factors, the current study has two objectives: (1) to estimate the prevalence of past-month nicotine vaping, nicotine-free vaping, and dual-use vaping (nicotine and nicotine-free) in high school students in Canada; and (2) to explore the correlations among past-month nicotine vaping, nicotine-free vaping, and dual-use vaping.

## 2. Materials and Methods

This study utilized data from the Canadian Student Tobacco, Alcohol and Drugs Survey (CSTADS), which was conducted between October 2018 and June 2019. The CSTADS utilized a stratified single-stage-cluster design to recruit students in grades 7 to 12 who attended private, public, or Catholic schools in the 10 Canadian provinces. In total, 116 school boards comprising 442 schools participated. The response rate at the student level was 69%. Students enrolled in school in the Yukon, the Northwest Territories, and Nunavut were excluded. Additionally, students enrolled in special needs schools, First Nations schools, virtual schools, or schools located on military bases were also excluded. The sample for the current study was restricted to students in grades 9 to 12 (or grades 9 to 11 in Quebec), as high school represents a period during adolescence that overlaps with important developmental milestones [22].

The primary outcome was past-month nicotine and nicotine-free vape use. Respondents were asked how often they had used a vape over the past 30 days, with responses measured on a 5-point scale (not at all, less than monthly, less than weekly, less than daily, daily, or almost daily). Since most respondents indicated no past-month use (73.8%), the variable was dichotomized into yes/no. A composite variable was then created to capture the mutually-exclusive categories of users, non-users, nicotine users, nicotine-free users, and dual (both nicotine and nicotine-free) users.

Demographic variables included sex (male/female), grade, urbanicity (urban/rural), and median household income at the school level (based on census tract areas from the 2016 Census). Health-related study variables included smoking status (current smoker, former smoker, non-smoker), lifetime alcohol use (yes/no), and lifetime cannabis use (yes/no).

### Statistical Analysis

Data were analyzed using Stata/SE 17.0. Percentages were used to summarize categorical variables. Multinomial logistic regression was used to assess correlations among past-month nicotine vaping, nicotine-free vaping, and dual-use vaping, with results presented as odds ratios and their corresponding 95% confidence intervals. Given the complex sampling design of the survey, sample weights and balanced repeated replication (BRR) with bootstrap weights were used to estimate variance in the regression model, which allowed for unbiased estimates once the properties of sample design were considered. Demographic and health-related variables were chosen based on their hypothesized relationships with vape use, as described in the wider literature. The final multinomial logistic regression model included the following covariates: sex, grade, urbanicity, median household income (at the school level), smoking status, lifetime alcohol use, and lifetime cannabis use. In total, 94% of the data were complete. Variables with missing data included vaping (3.8%), smoking status (0.2%), lifetime alcohol use (0.8%), and lifetime cannabis use (1.2%). A *p*-value < 0.05 was considered statistically significant.

## 3. Results

Table 1 compares baseline sociodemographic and health-related differences in the total sample (*N* = 38,229) and among nicotine and nicotine-free vapers. In total, 29.1% (*n* = 10,706) of the high school students indicated past-month vape use, with 12.2% (*n* = 4473) reporting vape use with nicotine and 2.8% (*n* = 1037) indicating vaping without nicotine. Approximately 14% (*n* = 5196) reported use of both nicotine and nicotine-free vapes in the past month.

Table 2 describes the weighted prevalence of vape use among grade 9-12 Canadian students. Overall, 2.5% of the students indicated nicotine-free-only vape use, 11.9% indicated nicotine-only vape use, and 11.3% indicated the use of both in the past 30-days.

Table 3 shows the weighted adjusted odds ratios for past-month nicotine vaping, nicotine-free vaping, and dual-use vaping (*N* = 36,766). For past-month nicotine vape use, males had a 39% increase in the odds of vape use compared to females (OR 1.39; 95% CI: 1.08, 1.79). Compared to students in grade 9, those in grade 10 had a 36% increase in the odds of nicotine vape use, whereas those in grade 11 had a 46% increase in the odds of nicotine vape use (OR 1.46; 95% CI: 1.09, 1.97). Compared to those who had never smoked, current smokers had a 379% increase in the odds of nicotine vape use (OR 4.79; 95% CI: 2.93, 7.83). Lifetime alcohol (OR 10.72; 95% CI: 7.15, 16.07) and cannabis use (OR 11.48; 95% CI: 9.85, 13.38) were associated with 972% and 1048% increases, compared to non-users, respectively, in the odds of nicotine vape use.

For past-month nicotine-free vape use, males had a 20% increase in the odds of nicotine-free vape use compared to females (OR 1.20; 95% CI: 1.01, 1.43). Compared to students in grade 9, students in grade 11 had a 44% decrease (OR 0.56; 95% CI: 0.38, 0.82) in the odds of past-month nicotine-free vape use. Lifetime cannabis use was associated with a 205% increase (OR 3.05; 95% CI: 2.47, 3.76) in past-month nicotine-free vape use.

For past-month dual-use vaping, males had a 31% increase in the odds of vape use compared to females (OR 1.31; 95% CI: 1.13, 1.53). Compared to students in grade 9, those in grade 11 had an 18% reduction in the odds of dual-use vaping, whereas those in grade 12 had a 51% decrease in the odds of past-month dual-use vaping (OR 0.49; 95% CI: 0.37, 0.64). Compared to those in the lowest category of median income, those with CAD 60,000–75,000 had a 39% increase in the odds of dual-use vaping (OR 1.39; 95% CI: 1.01, 1.92). Compared to those who had never smoked, current smokers had a 383% increase in the odds of dual-use vaping (OR 4.83; 95% CI: 3.15, 7.41). Lifetime alcohol (OR 9.10; 95% CI: 6.85, 12.08) and cannabis use (OR 9.10; 95% CI: 7.44, 11.13) were associated with an 810% increase, compared to non-users, in the odds of dual-use vaping.

## 4. Discussion

The estimated prevalence of high school vaping continues to be high (25.7%) and is consistent with other provincial and national estimates [5]. Approximately 12% of high school students reported vaping exclusively with nicotine vapes and 2.5% reported vaping exclusively with nicotine-free vapes in the past month. Additionally, and in contrast to other research, this study estimated the prevalence of dual-use vaping. Approximately 11.3% of high school students reported the use of both nicotine and nicotine-free vapes in the past month.

This study represents an exploration into sociodemographic and health-related differences between high school students who vape with and without nicotine. While similarities exist between the two types of users, several important differences were noted. In this study, males had higher odds than females of being in each category of past-month vape users. Other research noted similar findings but did not explicitly consider different categories of vaping behavior [23,24]. For example, in a subgroup analysis of university students who were not of legal age (<19 years) to purchase vaping products in Ontario, Canada, Seabrook et al. [7] found that cigarette smoking and cannabis use were the top factors associated with past-month vape use and that 21.8% of underage adolescents had vaped in the past month. The implication of this is that many underage youth are still able to access, and use vapes in Canada. In a large, non-probability school-based sample from Canada, Cole et al. [5] found that the prevalence of youth vaping increased across demographic groups before nicotine-containing vapes became legalized for sale in May 2018 and continued to increase thereafter. However, the study was not able to determine whether students were using vapes containing nicotine. The Canadian Tobacco and Nicotine Survey revealed that, among 15–19 year old Canadians in 2020, the most common reasons for vaping were for enjoyment (27%), just to try it (26%), and to reduce stress (23%) [8]. In a cross-sectional sample of 74,501 Canadian high school students from Alberta, British Columbia, Ontario, and Quebec, 39% of youth in 2018–2019 reported current substance use, 53% were using two or more substances, and vape use (28%) was the most prevalent of all substances and the most common substance combined with binge drinking [12]. Moreover, polysubstance use was more common among males, those with weekly spending money, Indigenous students, those experiencing depression, those having peer support, and those living in Alberta.

In the current study, age was significantly associated with both nicotine and nicotine-free vaping, but in opposite directions. Those in grades 10 and 11 were more likely than grade 9 students to be nicotine vape users, holding other variables constant, while those in grade 9 were more likely to be nicotine-free vape users compared to those in grade 11. For dual-use vaping, those in grade 9 were more likely to use both nicotine and nicotine-free vapes, compared with those in grade 11 or grade 12. Given that patterns of substance use tend to increase with age during adolescence, the nicotine-only results are not that surprising. However, the increased odds of nicotine-free and dual-use vaping among grade 9 students is concerning. Research has found that youth who vape nicotine-free products have a worse understanding of e-cigarette chemicals, compared with those who vape with nicotine [19]. Moreover, this lack of understanding might translate to the unknowing use, and to the subsequent misreporting, of nicotine vaping behavior [19,25]. In a cross-sectional study of four Connecticut high schools and two middle schools, Morean et al. [25] found that 34.1% of adolescents could not quantify the nicotine concentration of the e-liquid in their vapes, and that this was more common among females, nonsmokers, and those who vaped infrequently compared to nicotine users. The higher likelihood of dual-use vaping among grade 9 students in the current study warrants further investigation. Perhaps the behavioral and cultural pressures associated with transitioning to high school account for some of this variation, whereby vape use, in general, signals acceptance and belonging in high school culture. Alternatively, some of the associated mechanisms might be rooted in earlier (e.g., middle school) processes of socialization. Recent research suggests that both proximity and peer/media influence can have a substantial impact on the likelihood of initiating vape use [26,27]. Mantey et al. [26] found that observing vape use at school was associated with greater odds of becoming a user. This is especially concerning considering that they found that two out of three middle and high school students observed their peers vaping at school [26]. In addition to young students being exposed to high levels of vaping at school, Dai et al. [27], found that middle school students tend to be more susceptible to e-cigarette marketing and have lower levels of vaping media literacy, compared to high school students. Peer influence and stress relief were also key motivators for vape use among high school students in Toronto, Ontario [28].

Our findings regarding vaping and other substance use align with other research and add additional context to these associations. Indeed, other research has found that most individuals who smoke also vape [9]. Our findings corroborate this, with a caveat—smokers who vape tend to vape either with nicotine exclusively or with a mix of nicotine and nicotine-free, but rarely do they exclusively vape nicotine-free. These results are not surprising, given that youth see nicotine vape use as a strategy to limit cigarette smoking or to quit [29]. Lifetime alcohol and cannabis use were both associated with past-month vape use, with stronger associations for nicotine-vaping. This is consistent with other Canadian data showing that alcohol, tobacco, and cannabis were associated with a higher risk for vaping in adolescents, as was being employed and having lower grades in school [30]. Given that nicotine is commonly implicated in other patterns of polysubstance use, these results are not surprising [31,32].

Future research should continue to explore risk factors for different types of vape use. While nicotine and nicotine-free use represents one such “type”, other areas of vape use include cannabis and cannabis derivatives (e.g., oil, hash, shatter). Although nicotine vaping is more common than cannabis vaping, [33] one-third of adolescent vape users in Canada use their vaping device to consume cannabis [34]. This is concerning, since those who vape with THC and high-potency cannabis concentrates have a greater risk for psychosis, paranoia, and cannabis hyperemesis syndrome [33]. Thus, it is not surprising that the definition of vaping has expanded recently to include both “e-cigarettes” and the use of a vaporizer for cannabis [35]. Relatedly, vape use should not be regarded as a homogeneous activity and researchers should be explicit in their discussion of what type of vape use was measured. Second, research should focus on the more fundamental causes of vape use, such as the influence of socioeconomic status, gender, and race/ethnicity. This line of thinking is congruent with that proposed by Seabrook and Avison [36], whereby socioeconomic status, cumulative disadvantage processes, and health outcomes are assessed from a life-course perspective. They provided evidence, for example, showing how low socioeconomic status (e.g., living in poverty) is associated with higher risk-taking behavior (e.g., substance use) and greater exposure to and more kinds of stress, which in turn increases the risk of poor health outcomes, amplifying socioeconomic differences in health disparities over time. Consideration of the structural, socioeconomic, and demographic influences of vape use, as well as the development and implementation of surveillance tools that target socioeconomically disadvantaged populations, have also been recently advocated by the Canadian Public Health Association [21]. Similarly, future research should explore more temporal elements of vaping. Exploring patterns and transitions between types of vape use, longitudinally, would illuminate areas most amenable to intervention.

While the current study has several strengths, there are also limitations that need to be considered. Using survey data, the study followed a cross-sectional design, limiting the ability for causal inference [37]. The extent to which vaping behavior precedes, or follows, other health-related behaviors (alcohol use, marijuana use, smoking status) could not be established. Second, measures of vape and other substance use were all based on self-reports. Given the sensitive nature of this behavior, in conjunction with the young age range of the participants, many outcomes were likely underreported. Similarly, the contents of the vapes were self-reported and students may have been unaware of the nicotine content. However, self-reporting using validated instruments is the most common method to capture substance use [1]. Third, we could not establish any temporal sequence to the use of past-month vaping. Therefore, we were unable to differentiate between experimentation and current use or transitions between nicotine and nicotine-free use.

## 5. Conclusions

This study adds to existing research on youth vaping by establishing important distinctions between different types of vape users. As the prevalence of vape use has expanded, so too are the ways in which vapes are used. Our findings suggest that nicotine, nicotine-free, and dual-substance use remain a common pattern of vape use among Canadian high school students. The results of this study can aid future public health initiatives through the identification of key demographics for targeted initiatives. Greater awareness and programming around the dangers of nicotine-free vaping represents one such prospect. More work is also needed to prevent the initiation of vape use altogether and to deliver interventions that target polysubstance use among those who have started vaping.

## Figures and Tables

**Table 1 children-10-00368-t001:** Sample characteristics (overall and by vape status).

	No Vape Use	30-Day Nicotine Vape Use Only	30-Day Nicotine-Free Vape Use Only	30-Day Nicotine and Nicotine-Free Vape Use
Sex				
Female	13,249 (50.8%)	1934 (43.2%)	508 (48.9%)	2376 (45.7%)
Male	12,811 (49.2%)	2539 (56.8%)	529 (51.1%)	2820 (54.3%)
Grade				
9	8980 (34.4%)	662 (14.8%)	323 (31.1%)	1290 (24.8%)
10	7733 (29.7%)	1230 (27.5%)	312 (30.1%)	1573 (30.2%)
11	6069 (23.3%)	1411 (31.5%)	264 (25.5%)	1419 (27.3%)
12	3278 (12.6%)	1170 (26.2%)	138 (13.3%)	914 (17.6%)
Urban				
Urban	20,210 (77.6%)	3239 (72.4%)	763 (73.6%)	3734 (71.9)
Rural	5850 (22.4%)	1234 (27.6%)	274 (26.4%)	1462 (28.1%)
Median Income				
CAD 40,000–55,000	9448 (36.3%)	1185 (26.5%)	366 (35.3%)	1686 (32.5%)
CAD 60,000–75,000	9000 (34.5%)	1750 (39.1%)	410 (39.5%)	2063 (39.7)
CAD 80,000–120,000	7612 (29.2%)	1538 (34.4%)	261 (25.2%)	1447 (27.8%)
Smoking Status				
Non-Smoker	25,652 (98.5%)	3588 (80.8%)	995 (96.1%)	4329 (83.6%)
Former Smoker	113 (0.4%)	166 (3.7%)	8 (0.8%)	101 (2.0%)
Current Smoker	278 (1.1%)	688 (15.5%)	32 (3.1%)	746 (14.4%)
Lifetime Alcohol Use				
No	11,040 (42.5%)	144 (3.2%)	106 (10.2%)	217 (4.2%)
Yes	14,933 (57.5%)	4320 (96.8%)	930 (89.8%)	4958 (95.8%)
Lifetime Cannabis Use				
No	21,726 (84.1%)	982 (22.1%)	515 (50%)	1319 (25.7%)
Yes	4110 (15.9%)	3454 (77.9%)	515 (50%)	3824 (74.3%)
TOTAL	26,060 (70.9%)	4473 (12.2%)	1037 (2.8%)	5196 (14.1%)

**Table 2 children-10-00368-t002:** Weighted prevalence of vape use.

User Group	Prevalence	Standard Error	Confidence Interval
Non-User	74.3%	0.021	70.0–78.1%
Nicotine-Free User	2.5%	0.004	1.9–3.5%
Nicotine User	11.9%	0.013	9.5–14.7%
Nicotine and Nicotine-Free User	11.3%	0.008	9.8–13.0%

*N* = 36,766, population = 1,379,425.

**Table 3 children-10-00368-t003:** Adjusted logistic regression results: nicotine use, nicotine-free use, use of both.

			*N* = 36,274
	30-Day Vape with Nicotine	30-Day Vape without Nicotine	30-Day Vape with or without Nicotine
	Adjusted OR	Adjusted OR	Adjusted OR
Sex			
Female	1	1	1
Male	1.39 * [1.08, 1.79]	1.20 * [1.01, 1.43]	1.31 ** [1.13, 1.53]
Grade			
9	1	1	1
10	1.36 * [1.05, 1.77]	0.72 [0.46, 1.11]	0.96 [0.81, 1.14]
11	1.46* [1.09, 1.97]	0.56 * [0.38, 0.82]	0.82 * [0.67, 0.99]
12	1.34 [0.91, 1.97]	0.55 [0.30, 1.00]	0.49 ** [0.37, 0.64]
Urban			
Urban	1	1	1
Rural	1.23 [0.84, 1.80]	0.70 [0.43, 1.13]	1.01 [0.74, 1.39]
Median Income			
CAD 40,000–55,000	1	1	1
CAD 60,000–75,000	1.58 [0.96, 2.58]	1.52 [0.91, 2.53]	1.39 * [1.01, 1.92]
CAD 80,000–120,000	2.35 [1.28, 4.29]	0.69 [0.39, 1.20]	1.33 [0.84, 2.08]
Smoking Status			
Non-Smoker	1	1	1
Former Smoker	0.83 [0.27, 2.57]	0.58 [0.84, 3.00]	0.71 [0.22, 2.31]
Current Smoker	4.79 ** [2.93, 7.83]	1.38 [0.74, 2.59]	4.83 ** [3.15, 7.41]
Lifetime Alcohol Use			
No	1	1	1
Yes	10.72 ** [7.15, 16.07]	2.06 [0.82, 5.20]	9.10 ** [6.85, 12.08]
Lifetime Cannabis Use			
No	1	1	1
Yes	11.48 ** [9.85, 13.38]	3.05 ** [2.47, 3.76]	9.10 ** [7.44, 11.13]

Note: * indicates statistical significance at 0.05 level, ** indicates statistical significance at 0.01 level.

## Data Availability

Data available in a publicly accessible repository that does not issue DOIs. Publicly available datasets were analyzed in this study. Data were accessed on 1 November 2021. This data can be found here: [https://search1.odesi.ca/#/].

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
