# Peer review of "Nicotine and Nicotine-Free Vaping Behavior among a Sample of Canadian High School Students: A Cross-Sectional Study"

_children, 2023, doi:10.3390/children10020368_

Round 1

Reviewer 1 Report

Comments to authors:

The purpose of this study was to examine the prevalence of nicotine and nicotine-free vaping among Canadian high school students. Overall, the method is clear and appropriate, and the results are solid. I would like to point out the following as an attempt to further improve the manuscript contents:

1.      The introduction needs to be expanded. Please provide more information and literature review on high school students’ vaping behavior and the implications of understanding prevalence and correlates.

2.      Method session: lines 53-56, why these schools were excluded from this survey? Please provide reasons.

Author Response

The purpose of this study was to examine the prevalence of nicotine and nicotine-free vaping among Canadian high school students. Overall, the method is clear and appropriate, and the results are solid. I would like to point out the following as an attempt to further improve the manuscript contents:

Thanks so much for your comment.

  1. The introduction needs to be expanded. Please provide more information and literature review on high school students’ vaping behavior and the implications of understanding prevalence and correlates.

We have expanded our introduction and introduced additional research that explores high school student vaping behavior, prevalence, and correlates.

  1. Method session: lines 53-56, why these schools were excluded from this survey? Please provide reasons.

These schools were excluded from the survey by the survey creator, Propel Centre for Population Health Impact. We requested clarification from Propel but did not receive a response. However, it is likely they were excluded given the difficulties associated with active recruitment. The recruitment process involved contacting school boards first, in an attempt to capture all of the individual schools the boards governed.  

Reviewer 2 Report

The article investigates the prevalence and correlates of vaping behavior among adolescents in Canada. A cross-sectional survey is used.

The article is well-organized and focused.

These data complement prior reports of youth vaping that were investigated in other WEIRD nations. 

Most of my comments are minor. A few may be major depending on things that are not apparent to this reviewer. Comments are organized by section.

Introduction

1. The first sentence addresses nomenclature, which has varied and may continue to vary. Electronic cigarettes have at points and in populations been considered distinct from vapes. Would it be more consistent with the literature to use ENDS as the inclusive term?

2. The last sentence in the first paragraph implies a causal claim regarding the effect of vaping on cigarette smoking. Although this does seem to be reasonable and the confluence of evidence points in this direction. The cited papers don't present conclusive evidence on their own. The Audrain-McGovern/Leventhal analysis is prospective, they caution about making causal claims based on their data. It is unclear whether clarifying the rationale or softening the causal inference is appropriate.

Methods

3. Measurement. The key variable for this study is a measure of recent vape use. Respondents construct meaning from many components of the instrument and context. The face validity of the vape use item is potentially problematic. The item asks about the prior 30 days and the response options include not at all and less than monthly. Are respondents asked about lifetime use prior to this item?  The responses are set up to assess what is likely to elicit a schema rather than an enumeration of vaping events. Was this item piloted and validated? Is it known how infrequent users respond to this item?

Please comment on the potential for measurement error to impact the basic findings.

4. Please note what variables are in the multinomial regression. It is implied by the results tables but might benefit readers if it is explicit upfront.

5. Please note if cannabinoid vaping is assessed in the survey instrument. To what extent are respondents reporting using nicotine-free consumables when they are using THC/CBD, which might also have nicotine? This was briefly brought up in the discussion and a couple of important articles on this were cited. It wasn't clear to what extent this might be an issue for the current paper.

Results

6. Please report the number of primary sampling units.

7. Particularly for a paper that has the aim to address prevalence, it is valuable to include standard errors/CIs (e.g., Table 2). Total Survey Error and its components should also be addressed. What was the participation rate? What were the non-response rates? To what extent could these error generators influence the prevalence estimates?

8. It is not clear to me if it would be beneficial to report instead the row percentages rather than the column percentages in table 1. It does seem that if column percentages are retained, then normalization of the classes will benefit the reader. Of the never users, 12% are 12th graders, but twelfth graders are much less than a quarter of the sample.  Some readers may want the ease of access to other comparisons.

Author Response

The article investigates the prevalence and correlates of vaping behavior among adolescents in Canada. A cross-sectional survey is used. The article is well-organized and focused. These data complement prior reports of youth vaping that were investigated in other WEIRD nations. Most of my comments are minor. A few may be major depending on things that are not apparent to this reviewer. Comments are organized by section.

We thank the reviewer for these kind words.

Introduction

  1. The first sentence addresses nomenclature, which has varied and may continue to vary. Electronic cigarettes have at points and in populations been considered distinct from vapes. Would it be more consistent with the literature to use ENDS as the inclusive term?

Thank you for this comment. The literature certainly seems divided on this issue. Recent research, however, seems to favor ‘vaping’ or ‘e-cigarette’, especially for studies investigating both nicotine and nicotine-free use. Given the implicit association between e-cigarettes and nicotine we have decided to use the more neutral language of ‘vaping’.

  1. The last sentence in the first paragraph implies a causal claim regarding the effect of vaping on cigarette smoking. Although this does seem to be reasonable and the confluence of evidence points in this direction. The cited papers don't present conclusive evidence on their own. The Audrain-McGovern/Leventhal analysis is prospective, they caution about making causal claims based on their data. It is unclear whether clarifying the rationale or softening the causal inference is appropriate.

Thank you for this comment. We agree that the inference made was too strong. We have described and clarified the evidence more thoroughly and articulated how adolescent vape use is correlated with initiating cigarette smoking.

Methods

  1. Measurement. The key variable for this study is a measure of recent vape use. Respondents construct meaning from many components of the instrument and context. The face validity of the vape use item is potentially problematic. The item asks about the prior 30 days and the response options include not at all and less than monthly. Are respondents asked about lifetime use prior to this item?  The responses are set up to assess what is likely to elicit a schema rather than an enumeration of vaping events. Was this item piloted and validated? Is it known how infrequent users respond to this item?

Please comment on the potential for measurement error to impact the basic findings.

That is a fair comment. However, respondents were not asked about lifetime use prior to this question. The authors agree that the question provides a schema but do not think this precludes an aggregated enumeration. On the actual survey the categories of vape use were thoroughly described (e.g., the weekly category was presented as “< daily but at least once a week”). The survey was also piloted twice and included focus groups immediately thereafter. Given the detail provided for each response category, we believe placement within categories was straightforward. In this regard, the potential for measurement error is limited to biases associated with all substance use research, namely, social desirability bias.  

  1. Please note what variables are in the multinomial regression. It is implied by the results tables but might benefit readers if it is explicit upfront.

That is an excellent point. We have added the variables to the statistical analysis subsection.

  1. Please note if cannabinoid vaping is assessed in the survey instrument. To what extent are respondents reporting using nicotine-free consumables when they are using THC/CBD, which might also have nicotine? This was briefly brought up in the discussion and a couple of important articles on this were cited. It wasn't clear to what extent this might be an issue for the current paper.

The survey instrument assessed cannabis vaping in another section. We did not use that section given low response rates, coupled with the fact that it was only measured in reference to the preceding 12 months. The questions used for vaping came from the tobacco section and were limited to e-cigarettes with nicotine and e-cigarettes that were nicotine-free.

Results

  1. Please report the number of primary sampling units.

Thank you. We have added this to the manuscript. In total, 116 school boards with 442 schools participated.

  1. Particularly for a paper that has the aim to address prevalence, it is valuable to include standard errors/CIs (e.g., Table 2). Total Survey Error and its components should also be addressed. What was the participation rate? What were the non-response rates? To what extent could these error generators influence the prevalence estimates?

We have added standard errors and confidence intervals to Table 2. We have also added the response rate to the Methods section. In total, 69% of eligible students participated. Non-response was a function of parental refusal, student refusal, and absenteeism. Given the many reasons for non-response, it is difficult to assess the extent to which it could influence the prevalence estimate, although some level of differential non-response, especially amongst the ‘student refusal’ group is likely.  

  1. It is not clear to me if it would be beneficial to report instead the row percentages rather than the column percentages in table 1. It does seem that if column percentages are retained, then normalization of the classes will benefit the reader. Of the never users, 12% are 12th graders, but twelfth graders are much less than a quarter of the sample.  Some readers may want the ease of access to other comparisons.

Thank you for this suggestion. Intuitively, we prefer to keep the column percentages. Although normalization is possible, we do not think it would improve readability.